# Highly efficient serum-free manipulation of miRNA in human NK cells without loss of viability or phenotypic alterations is accomplished with TransIT-TKO

Breanna K. V. Hargreaves[1], Sarah E. Roberts[2], Beata Derfalvi[1,2☉], Jeanette E. Boudreau[1,3☉] *

1 Department of Microbiology and Immunology, Dalhousie University, Halifax, Canada, 2 Department of Pediatrics, Dalhousie University, Halifax, Canada, 3 Department of Pathology, Dalhousie University, Halifax, Canada

☉ These authors contributed equally to this work.
* Jeanette.boudreau@dal.ca

**Data Availability Statement:** All relevant data are contained within the manuscript and supplemental files.

## Abstract

Natural killer (NK) cells are innate lymphocytes with functions that include target cell killing, inflammation and regulation. NK cells integrate incoming activating and inhibitory signals through an array of germline-encoded receptors to gauge the health of neighbouring cells. The reactive potential of NK cells is influenced by microRNA (miRNA), small non-coding sequences that interfere with mRNA expression. miRNAs are highly conserved between species, and a single miRNA can have hundreds to thousands of targets and influence entire cellular programs. Two miRNA species, miR-155-5p and miR-146a-5p are known to be important in controlling NK cell function, but research to best understand the impacts of miRNA species within NK cells has been bottlenecked by a lack of techniques for altering miRNA concentrations efficiently and without off-target effects. Here, we describe a non-viral and straightforward approach for increasing or decreasing expression of miRNA in primary human NK cells. We achieve >90% transfection efficiency without off-target impacts on NK cell viability, education, phenotype or function. This opens the opportunity to study and manipulate NK cell miRNA profiles and their impacts on NK cellular programs which may influence outcomes of cancer, inflammation and autoimmunity.

## Introduction

Natural killer (NK) cells are potent effectors for maintenance of immune homeostasis, with influences in successful pregnancy, control of autoimmunity, and protection against cancer and viral infections. NK cell functions range from regulation to inflammation, and include cytokine production and cytotoxicity through direct binding and degranulation, promotion of apoptosis and antibody-dependent cellular cytotoxicity (ADCC) [1, 2].

**Funding:** This work was supported by studentships to BKVH from the Nova Scotia Graduate Scholarship and the Nova Scotia Health Research (NSHRF) foundation, and grants to BD (Canada Foundation for Innovation, John R. Evans Leaders Fund #34283, NSHRF Establishment Grant #1045, IWK Establishment Grant 2017, Dalhousie University Dept. of Pediatrics Endowment Fund, 2018), the Natural Sciences and Engineering Research Council #06489 of Canada to JEB, and Canadian Institutes of Health Research (CIHR), REACH Program Team Grant: Health Challenges in Chronic Inflammation Initiative (CIHR 135230). The sponsors did not play a role in any aspect of the study.

**Competing interests:** The authors declare that no competing interests exist.

The outcome of an NK-target interaction is determined by the NK cells' integration of simultaneous signaling through germline-encoded activating and inhibitory receptors. Receptor expression is not universal across all of the NK cells in an individual's repertoire: up to 30,000 phenotypically-distinct NK cell subsets have been defined in a single donor [3]. The functional potential of individual NK cells is calibrated by their capacity for inhibition by "self" human leukocyte antigens (HLA) in a process called "education" or "tuning" [1, 4, 5]. This creates a repertoire of NK cells in each individual comprised of NK cells differently equipped to respond to immunologic challenges [1, 6, 7]. In response to certain viruses and haptens, NK cells can form memory-like or "adaptive" populations that are long-lived and exhibit enhanced responsiveness upon rechallenge [8–10]. NK cells are highly plastic, and have clear potential for the immunotherapy of cancer, infectious disease and autoimmune disorders, but further study is needed to understand how to best and broadly control NK cell responsiveness.

At rest, and especially in response to cytokine pre-priming, NK cells encode an array of pre-formed mRNA for effector molecules including cytokines, granzymes and signaling receptors. MicroRNAs (miRNAs) can bind to and suppress mRNA translation; this enables post-transcriptional control for dynamic modulation of NK cell function [11]. MiRNAs are found throughout body fluids and cells, evolutionarily conserved and highly stable [12]. A single miRNA can have hundreds of targets, so miRNA enable modulation of cellular processes by impacting an array of mRNA species [13]. As a result, miRNAs have multifactorial impacts on human health and, when dysregulated, can underlie autoimmune diseases, antiviral immunity and cancer [14, 15].

The broad regulation of cellular processes by miRNA make them intriguing as agents to modulate NK cells for immunotherapy and to monitor ongoing processes. Already, miRNA species have been implicated as biomarkers for identifying and characterizing diseases or monitoring treatment efficacy [16–18]. Noteworthy, one miRNA-based treatment, miravirsen, is being tested against hepatitis C (clinicaltrials.gov # NCT02508090). Miravirsen is an antisense oligonucleotide that interferes with miR-122, a miRNA that otherwise facilitates persistence of viral RNA [19]. This trial proves the principle that it is feasible to deliver, manipulate and measure miRNA species, for clinical applications. As central mediators of inflammation and regulation and with functions controlled by miRNA, NK cells are an ideal target population for miRNA-based therapies, but further studies are required to understand how miRNA impact NK cell function.

Approaches are established for expansion, engineering and delivery of NK cells [20–22], but strategies to efficiently target miRNA in particular are lacking. Tailoring of the miRNA environment in NK cells may allow control in complex cellular processes to polarize NK cells for inflammation or regulation. As with many cell subtypes, transfection and engineering of primary NK cells has been challenging, and can induce alterations in NK cell biology and function as the NK cell responds to the transfection process or reagents [23]. These off-target impacts may alter NK cell function, with consequences on the efficacy of the desired intervention or interpretation of experiments. Few studies have altered miRNA in primary human NK cells, possibly owing to a lack of efficient methods for this purpose. Non-viral methods include lipid-based (lipofectamine, HiPerfect) [24–26] or mechanical (i.e. electroporation, nucleofection) [27]. A limited array of miRNA have been assessed using these methods, and the highest reported transfection efficiencies are 35% [26]. Noteworthy, among all published studies, cells were maintained in serum-enriched media. For investigations of miRNA in particular, it is important to avoid introduction of highly-conserved exogenous miRNA from serum [28], necessitating that all miRNA manipulations must be conducted in serum-free media. Approaches that enable manipulation of miRNA without substantially altering NK cell

education, phenotype or memory in serum-free media are therefore needed to better understand the impacts of miRNA on NK cell function.

Aiming to develop a technique to efficiently engineer NK cells through manipulation of miRNA, we tested four approaches. We modified expression of two miRNA species with known links to both human health and NK cell function: miR-155-5p and miR-146a-5p and tested their ability to change the miRNA landscape without altering population variation, NK cell education and hallmarks of NK cell adaptation. We found that the TransIT-TKO transfection reagent–a tool originally designed to deliver siRNA into cells–far outperformed other transfection approaches. Using fluorescein (FAM)-labeled miRNA, we demonstrate that >90% of unmanipulated, primary human NK cells can be transfected in this way, with maintenance of >80% viability. NK cell phenotype, repertoire diversity and education remained intact following transfection, confirming that this approach is highly-effective for engineering of NK cells without off-target impacts. To our knowledge, this represents the highest efficiency transfection of primary NK cells. We expect that this simple approach will accelerate miRNA engineering in NK cells for their use in immunotherapy and discovery research.

## Methods

### Ethics

Peripheral blood was collected from healthy donors locally and in collaboration with Canadian Blood Services' Blood4Research program. Fibroblast-like synoviocytes (FLS) were collected from juvenile idiopathic arthritis (JIA) patients. The study was approved by the research ethics boards of the Dalhousie University, IWK Health Center and Canadian Blood Services and all participants or their guardians signed informed consent.

### Cell isolation, cell lines and culture

PBMCs were isolated by density gradient centrifugation and NK cells were isolated from fresh heparinized blood sample using the RosetteSep Human NK Cell Enrichment Cocktail (Stemcell Technologies), according to the manufacturer's instructions. We compared ATCC recommended media (containing 25% bovine and horse serum) and RPMI media (containing 10% FBS) to serum-free X-VIVO 10 media (Lonza) to define conditions for cell cultures without serum. Unless otherwise indicated, primary PBMCs and NK cells were maintained in X-VIVO 10 Serum-free Hematopoietic Media containing L-glutamine (3mM, Gibco), L-serine (1.8mM, Alfa Aesar), L-asparagine (0.6mM, Multicell), and 100 IU/mL IL-2 (Peprotech) at 1-$2x10^6$ cells/mL at 37°C and 5% $CO_2$. Previously frozen PBMCs were rested overnight and NK cells rested for a minimum of 2 hours prior to experimental manipulation(s).

FLS were isolated from whole synovial fluid collected from patients with polyarticular or oligoarticular JIA using established protocols developed with modifications [29]. Briefly, cells were pelleted from whole synovial fluids (1000x$g$, 15 min) and mononuclear cells were isolated by density gradient centrifugation. Synovial Fluid MNCs were plated for 24 hours and non-adherent cells were removed. Adherent cells were maintained in alpha-MEM containing 20% HI FBS, NEAA, 1 μM sodium pyruvate, penicillin/streptomycin, and 2 μM L-glutamine (all from Gibco). Once confluent, we confirmed that FLS cell lines were homogeneous by flow cytometry phenotyping: CD14-negative, CD90 positive, Cadherin 11 positive. Thereafter, cells were used for experimentation at passage four and transferred to serum-free media, (X-Vivo 10) 24 hours prior to transfection.

The NK cell line, NK-92 (ATCC), was maintained in ATCC recommended media and cultured in X-VIVO 10 for experiments. The K562 target cell line was obtained from the National Cancer Institute and maintained in RPMI with 10% FBS.

## Transfection

To identify the ideal procedure for manipulating miRNA in human NK cells, we tested several approaches, including the Amaxa Human NK Cell Nucleofector Kit (Lonza), Lipofectamine RNAiMAX Transfection Reagent (ThermoFisher Scientific), TransIT-SiQuest Transfection Reagent (Mirus), and TransIT-TKO Transfection Reagent (Mirus), all following the manufacturer's recommended protocols. All optimization transfections for miRNAs were completed with 25 nM of FAM-labeled Negative Control miRCURY LNA miRNA Mimic (Qiagen) in X-VIVO media compared to reagent only and no transfection controls. For all transfections, viability was assessed by trypan blue exclusion and flow cytometry, and efficiency was determined by flow cytometry. Based on these results, TransIT-TKO was optimized for primary human NK cell transfection with sense and anti-sense miRNAs.

Briefly, $1 \times 10^6$ NK cells were resuspended in 510 μL X-VIVO media containing 100 IU/mL IL-2. In a separate sterile, RNase free, microfuge tube, 50 μL Opti-MEM Reduced serum media (ThermoFisher Scientific) was combined with 1 μL TKO reagent and 25 nM FAM-labelled negative control, mimic (sense) or antisense miRNA (Qiagen) and incubated in the dark for 20 minutes. The Opti-MEM-TKO-miRNA solution was added dropwise to the cells and the plate was rocked gently. Transfected NK cells were incubated for a minimum of 18 hours and maximum of 4 days at 37°C under 5% $CO_2$ conditions.

## Molecular biology

Mimic (sense) and antisense (inhibitors) for miRNA or a negative control mimic were obtained from Qiagen (miRCURY LNA miRNA Mimics/Inhibitors with FAM label, **S1 Table**). To confirm and quantify sense and antisense miRNA delivery post NK cell transfection, miRNA expression was assessed by real time quantitative PCR (RTqPCR). Total RNA, including miRNA, was isolated using the miRNeasy Mini Kit including an on-column DNase digestion (Qiagen) as described by the manufacturer. RNA was tested for quantity and quality using a NanoDrop spectrophotometer prior to cDNA synthesis. To create cDNA, a miScript II RT kit (Qiagen) was used according to manufacturer's instructions. The final cDNA template was diluted 1/10 and stored at -20°C. To confirm that there was no DNA contamination, all cDNA reactions included at least one reaction that omitted the reverse-transcriptase, (no RT control), and this cDNA was run for all miRNA targets and reference genes to confirm that no PCR product was made. RTqPCR reaction mixtures contained molecular grade sterile water, 1x SsoAdvanced Universal SYBR Green Supermix, 0.25 mM forward and universal reverse primers, and 5 μL of the appropriate cDNA template. Reference genes, *miR-103a-3p* and *miR-191-5p*, were selected by GeNorm algorithm [30, 31]. RTqPCR cycles (**S2 Table**) were completed with CFX Connect real-time PCR detection system (BioRad) and analyzed with CFX Maestro software. All primer sequences and annealing temperatures can be found in **S3 and S4 Tables**.

## Flow cytometry

To assess possible phenotypic and functional changes to the cells, NK cells transfected with the negative control miRNA mimic were stained and assessed by flow cytometry. Antibodies, clones and concentrations used are shown in **S4 Table**. Non-specific binding was prevented by incubation with 1.3 mg/mL Fc Block containing a mixture, (1:1) of human IgG and heat aggregated IgG, (CSL Behring) at 4°C. After 15 minutes, surface stains, prepared in brilliant stain buffer (BD Bioscience), were added directly to the cells, and the cells were incubated in the dark for a further 20 minutes. The cells were washed with 1x PBS and stained with eBioscience fixable viability dye (ThermoFisher) for 30 minutes at 4°C then washed in staining buffer (5

min, 700 x *g*). In some experiments, the cells were stained intracellularly using the Fixation/ Permeabilization Solution kit (BD Bioscience) according to the manufacturer's instructions. Following staining, cells were resuspended in 1% paraformaldehyde for fixation. Flow cytometers in the Dalhousie University's CORE facilities (BD FACS Canto II or BD Fortessa with BD FACSDIVA software), were used for all flow cytometry data collection and data were analyzed using FlowJo 10.5 software.

### Degranulation and cytotoxicity assays

To assess possible changes in NK cell function after TransIT-TKO transfection, we performed cytotoxicity assays using K562 to measure direct killing [32], and Rituximab (RTX)-coated autologous B cells to measure ADCC [33]. Prior to each assay, target cells (K562 or RTX-coated B cells) were stained with cell trace violet (CTV) (Thermofisher). Negative control mimic transfected and non-transfected NK cells were co-cultured with CTV-stained target cells such that the NK to target cell ratio was 1:1, and centrifuged at 200 x *g* for 3 minutes to promote cell-cell contact. K562 co-cultures were incubated for 5 hours and autologous PBMC co-cultures were incubated for 2 hours with or without 5 μg/mL RTX. Both co-cultures were maintained in X-VIVO 10 media with anti-LAMP1 (CD107a) antibody. To assess functional results (cytotoxicity and degranulation) co-cultures were stained for flow cytometry analysis.

### Statistical analysis

All statistical analyses were conducted on either normalized RTqPCR relative gene expression or flow cytometry geometric means as appropriate. Samples were tested for normality with the Shapiro-Wilk normality test, and passed normality if $\alpha = 0.05$. If the data passed normality, analysis of variance (ANOVA) and parametric matched ratio paired *t* tests were completed. If the data did not pass normality, paired non-parametric Wilcoxon tests were performed. Data for all statistical tests was deemed significant if $p < 0.05$.

## Results

### Establishment of serum-free growth conditions for primary human NK cells

MiRNAs are extremely conserved, often having exact or highly homologous sequences across mammalian species. We compared the sequences for miR-155-5p and miR-146a-5p between humans, cows, horses and mice: species whose serum is most often used in the culture of human NK cells. As expected, there is extensive inter-species conservation for these miRNA (S5 Table). To avoid introduction of extraneous miRNAs through culture and/or transfection, we developed serum-free culture conditions for primary human NK cells.

NK-92 and primary human NK cells were cultured for up to four days, and cellular viability was assessed by trypan blue exclusion and flow cytometry (S1 Fig). NK-92 cells grown in X-VIVO and RPMI maintained a viability of 95% but cells grown in ATCC recommended media exhibited a decreased viability of 85% after four days. Surprisingly, primary NK cells grown in X-VIVO media maintained a higher viability (92±2%) than those cultured in ATCC media (87±7%) after four days of culture. Cellular viabilities did not significantly differ between the ATCC recommended media for the NK-92 cell line or primary human NK cells and all subsequent experimentation was therefore conducted using serum free X-VIVO 10 media.

## TransIT-TKO outcompetes other transfection techniques for delivering sense and antisense miRNAs to primary human NK cells

To determine the best technique for primary NK cell transfections, we compared the efficiency and viability of multiple transfection techniques, including lipofectamine, nucleofection, TransIT-SiQuest, and TransIT-TKO, a reagent created for delivery of siRNA (**Fig 1**). We used a fluorescein (FAM)-labeled control miRNA which encodes only a "scramble" sequence (i.e. no specific miRNA) to compare transfection approaches. The FAM label was included in this and all transfections (control, mimic and antisense). FAM allowed us to track transfection efficiency as the proportion of FAM+ among viable NK cells after transfection, and persistence of labeled oligonucleotides. Among the transfection methods tested, only TransIT-TKO could introduce miRNA efficiently (93.4+/-2.9% transfected), and without substantial mortality among recipient cells (93±2.8% viable at 2.5 days post-transfection). Similar transfection efficiencies were obtained for scramble oligonucleotide controls and all miRNA mimics and inhibitors used in subsequent experiments. To our knowledge, this is the most efficient transfection of NK cells reported.

We were surprised by the high and consistent miRNA NK cell transfection efficiencies without compromise to cellular viabilities achieved by TransIT-TKO. To determine whether this reagent was universally effective for delivery of miRNA, we tested its utility for transfecting FLS and primary PBMCs (**S2 Fig**). Between all three cell types we achieved an average viability of 94.2±5% and efficiency of 98.4±0.3%. TransIT-TKO successfully transfected both hematopoietic and non-hematopoietic cells, indicating its universal utility for delivery of mimic or antisense miRNA species.

## miRNA-155-5p and miRNA-146a-5p expression can be controlled by transfection with TransIT-TKO

We next sought to alter the miRNA microenvironment in NK cells by introducing mimic miRNA or antisense miRNA to increase or silence expression of specific miRNA species, respectively. For this experiment, we elected to alter expression of miRNA-155-5p or miR-146a-5p, two miRNA species known to impact NK cell function [11, 34]. Following transfection with sense miRNA-155-5p or miR-146a-5p, we observed ~1000-fold and sustained increases in expression of these miRNA species in primary human NK cells (**Fig 2A and 2B**). Persistent expression of the transfectant-encoded FAM protein was observed in over 90% of cells by flow cytometry and similar between cells transfected with scramble or miRNA-modifying oligonucleotides. Furthermore, RT-qPCR revealed quantitative changes in the expression of intracellular miRNAs: delivery of the antisense miRNAs led to a significant decrease in the quantity of miRNA-146a-5p transcripts (0.3±0.08, p = 0.0288) and a trend toward decreased miRNA-155-5p (0.23±0.28) in transfected cells.

To determine whether these changes could impact miRNA targets, we measured STAT-1 and IRAK-1 mRNA expression in cells after transfection with miRNA-146-5p mimic or scrambled control oligonucleotide. As predicted, both STAT-1 and IRAK-1, direct targets of miRNA-146-5p [35, 36], were diminished 18h after transfection (Fig 2C). Taken together, these results reveal that miRNA species can be quickly manipulated in primary human NK cells.

## The distribution of NK cell subsets is not altered by miRNA transfection

To study the impacts of miRNA on NK cell function, a technique that can alter miRNA while preserving normal NK cell function and phenotype is needed. We compared NK cells

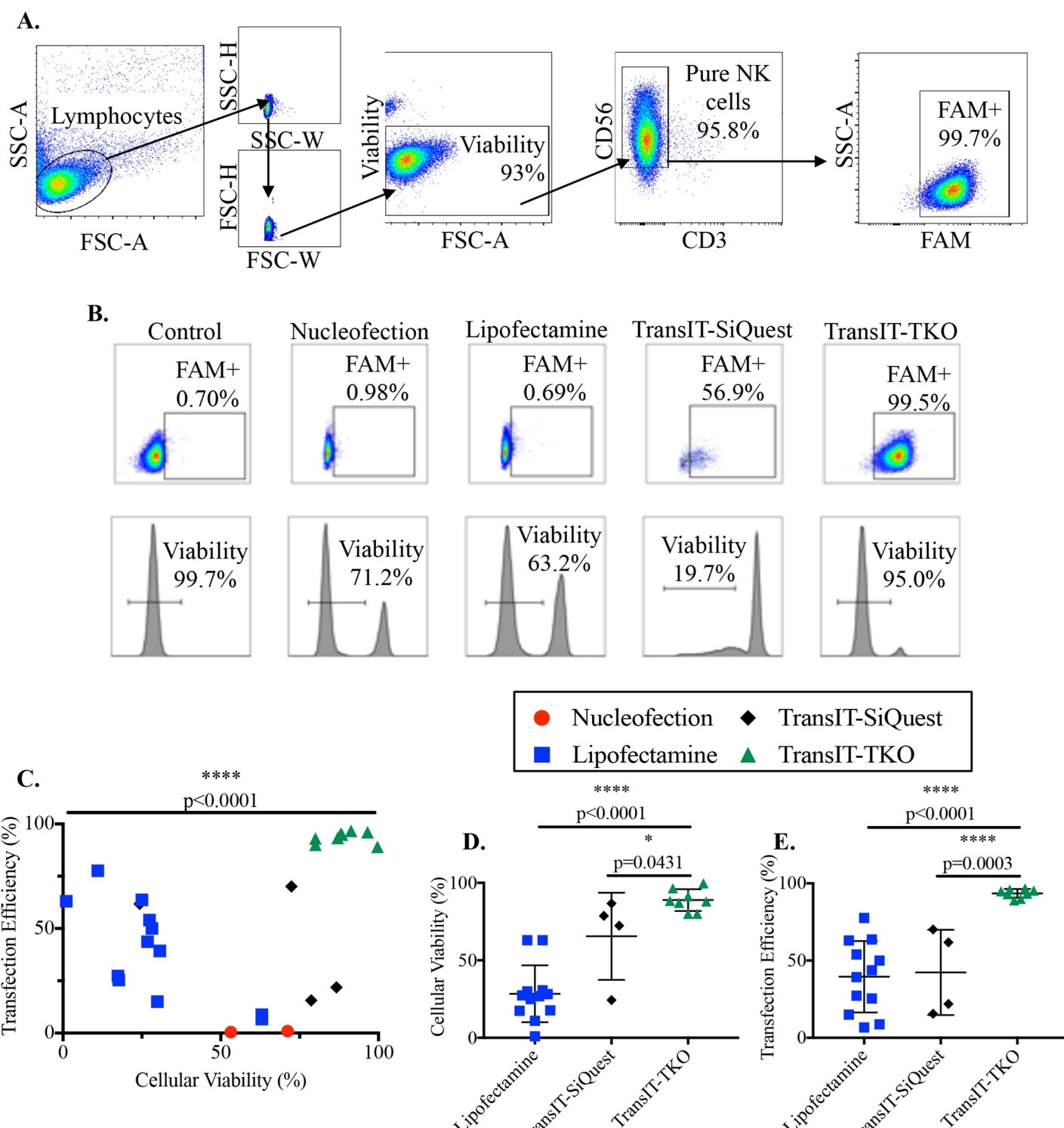

**Fig 1. TransIT-TKO outcompetes other transfection techniques.** RosetteSep isolated primary human NK cells were transfected with 25 nM FAM-labeled negative control (scramble oligonucleotide) for 24 hours using nucleofection (red circle), lipofectamine (blue square), TransIT-SiQuest (black diamonds), or TransIT-TKO (green triangles). A-B) Cellular purity, viability, and transfection efficiency was assessed by flow cytometry. C-E) Comparison of transfection efficiencies and viabilities between transfection techniques. Plots represent individual transfection attempts, bars represent mean ± standard deviation. Data was assessed by one-way ANOVA (C) and unpaired *t* tests (D-E).

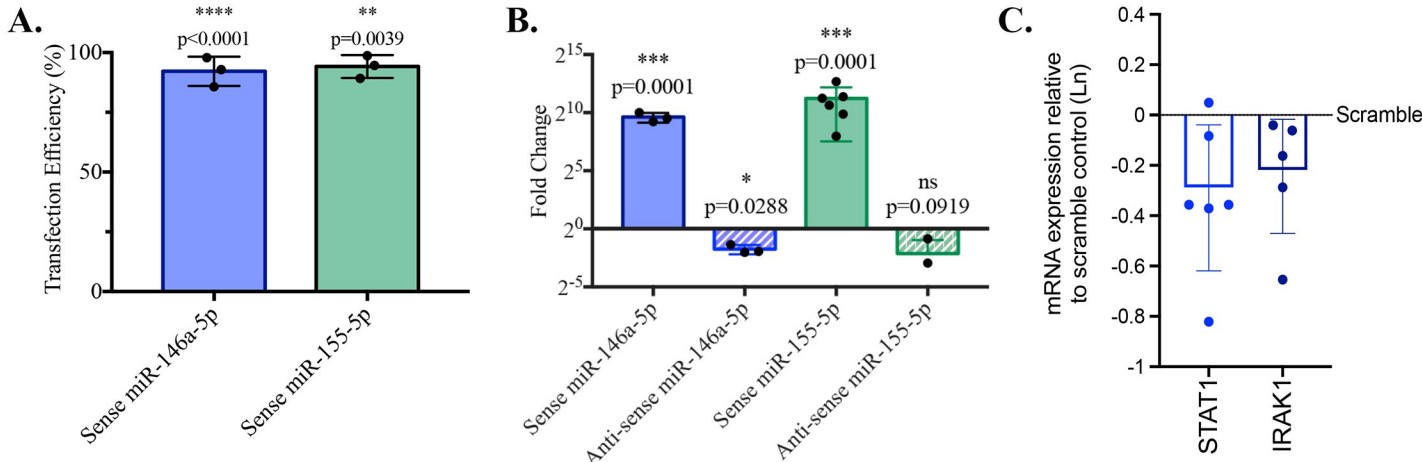

**Fig 2. TransIT-TKO delivery of sense and antisense miRNAs alters miRNA NK landscape.** RosetteSep isolated primary human NK cells were transfected with 25 nM miR-146a-5p or miR-155-5p sense or antisense miRNA using TransIT-TKO. A) Sense miRNA transfection efficiencies were determined by flow cytometry to measure FAM. B) MiRNA sense and antisense delivery was measured by RTqPCR and compared to negative control miRNA. Reference miRNAs included miR-103a-3p and miR-191-5p. Data represents individual transfections (efficiency or fold change) and bars represent mean ± standard deviation, n = 3–4. RTqPCR results were assessed by one-way ratio paired *t* tests. C) mRNA for two targets of mR-146a-5p, STAT1 and IRAK1, was measured 18h post transfection. Values are presented as fold change compared with mRNA values from the same donor after transfection with scramble oligonucleotide.

transfected with a non-specific negative control mimic to non-transfected cells to determine whether TransIT-TKO or the introduction of exogenous miRNA nonspecifically influences the phenotype of NK cells. After 2.5 days, we measured and compared the phenotype of NK cells from the same donor after each treatment (**Fig 3**). Recognizing the extensive donor to donor variation that occurs, we compared the distribution of NK cell subsets before and after transfection with matching based on donor. Our first flow cytometry panel tested the distribution of the inhibitory KIR and NKG2A receptors, which are known to be important for NK cell education (**Fig 3A**). In our second panel, we queried markers of adaptive/memory NK cell populations (**Fig 3B**). Neither phenotypic analysis exhibited substantial changes in response to transfection, indicating that this highly-efficient technique for miRNA manipulation does not skew NK cell population frequencies.

## NK cell cytotoxic function by missing self-responsiveness and ADCC are not impacted by miRNA transfection

To ascertain whether transfection changed the reactive function of NK cells, we tested whether missing self-responsiveness was altered by transfection. We tested populations from the same donor with and without stimulation using the HLA-negative NK cell target K562 (**Fig 4**). As expected, NK cells responded to the "missing ligand" target, K562 with increased IFN-γ production and degranulation (CD107a externalization) compared to unstimulated cells. Consistent with previous observations [37], granzyme B was depleted after degranulation of NK cells. Importantly, the introduction of negative control miRNA mimics did not alter the magnitude of this missing self-reactivity among NK cells. Hence, TransIT-TKO transfection can be used to alter the miRNA landscape without non-specifically impacting the fundamental physiology of NK cells.

A major feature of NK cells is their ability to mediate antibody-dependent cellular cytotoxicity (ADCC). We tested NK cell-mediated ADCC by incubating NK cells with and without transfection of negative control miRNA mimics with autologous B cells in the presence of anti-CD20 (Rituximab, RTX). Similar to missing self-responsiveness, NK-mediated ADCC

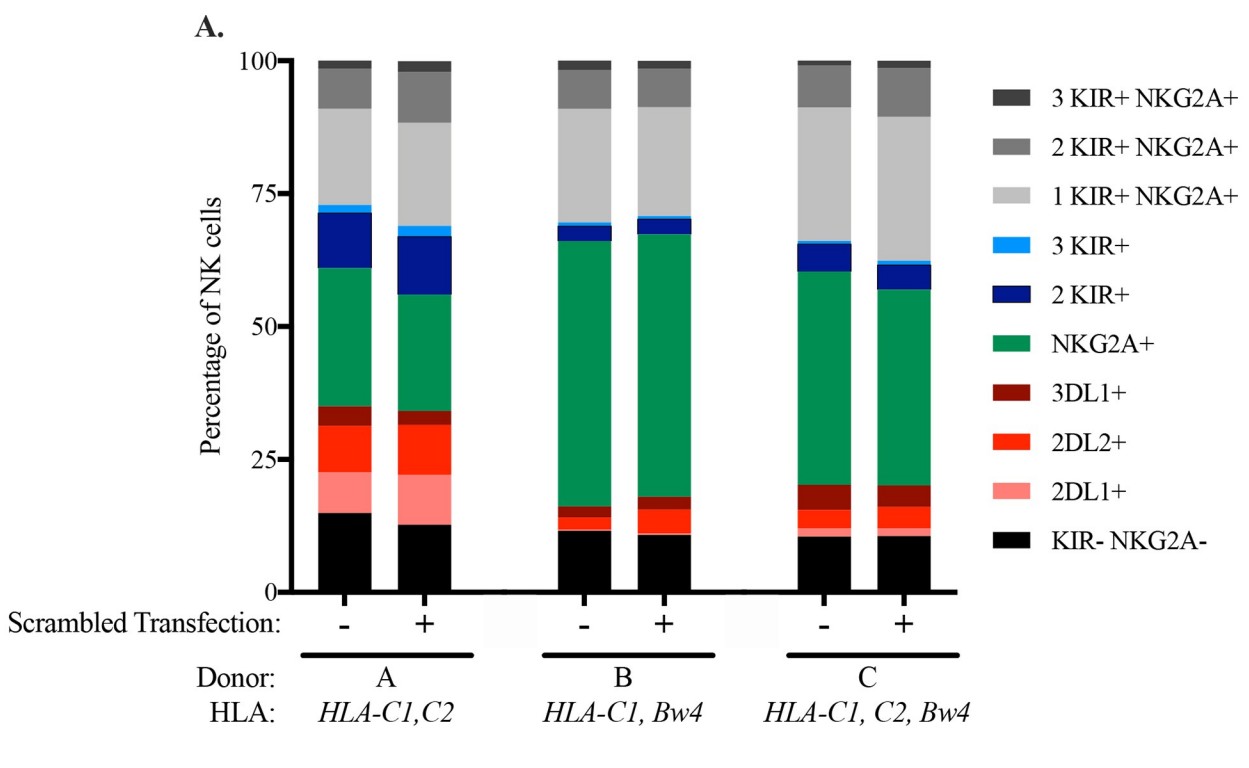

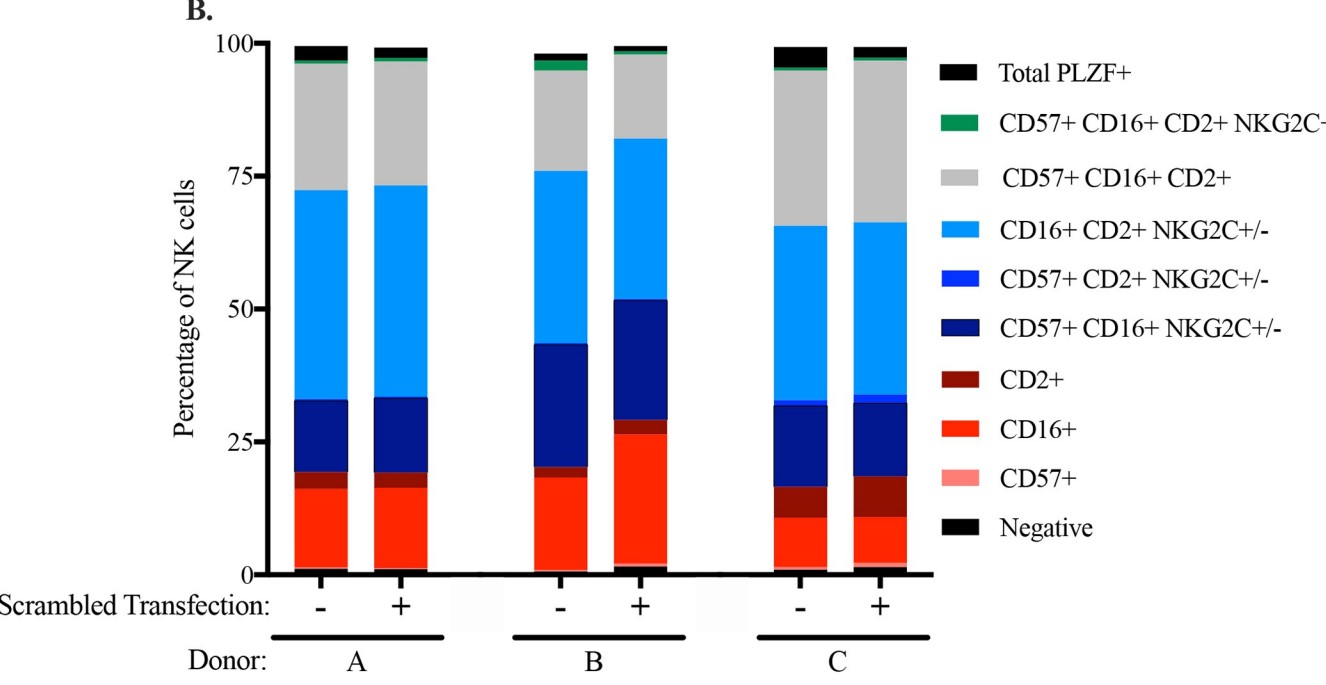

**Fig 3. TransIT-TKO transfection with negative control mimic does not impact NK cell phenotypes and distribution.** RosetteSep isolated human NK cells were transfected with Negative Control Mimic for 2.5 days with TransIT-TKO. NK cell phenotypic markers were assessed by flow cytometry and donors were genotyped for *HLA*.

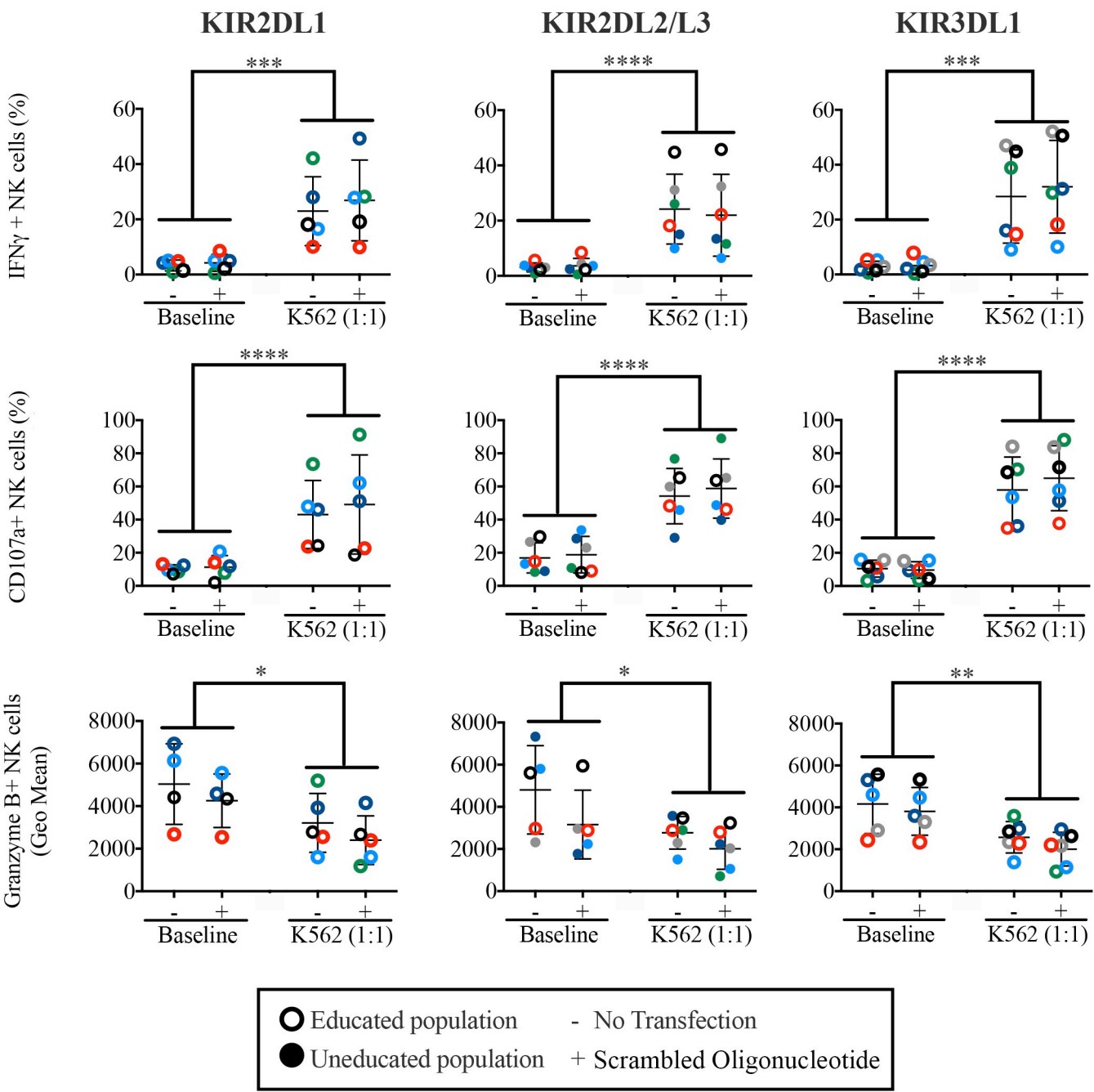

**Fig 4. Missing self reactivity is not impacted by miRNA transfection.** NK cells were isolated form healthy human donors and transfected for 2.5 days using a negative control miRNA mimic, or mock transfected. Thereafter, NK cells were challenged with the HLA-negative target K562 and IFN-γ and degranulation were measured. NK cells predicted to be educated based on donor HLA are shown as hollow circles; NK cells predicted to be uneducated are shown as filled symbols. Each donor is represented by a unique color.

was not altered by TransIT-TKO transfection or the introduction of non-targeting miRNA mimics (**Fig 5**). Taken together, these functional studies confirm that the impacts of miRNA sense or antisense strands to NK cells is highly effective and does not lead to off-target impacts on NK cells. Therefore, this approach to alter the NK miRNA microenvironment will be useful in determining the specific impacts of miRNA species on NK cell function.

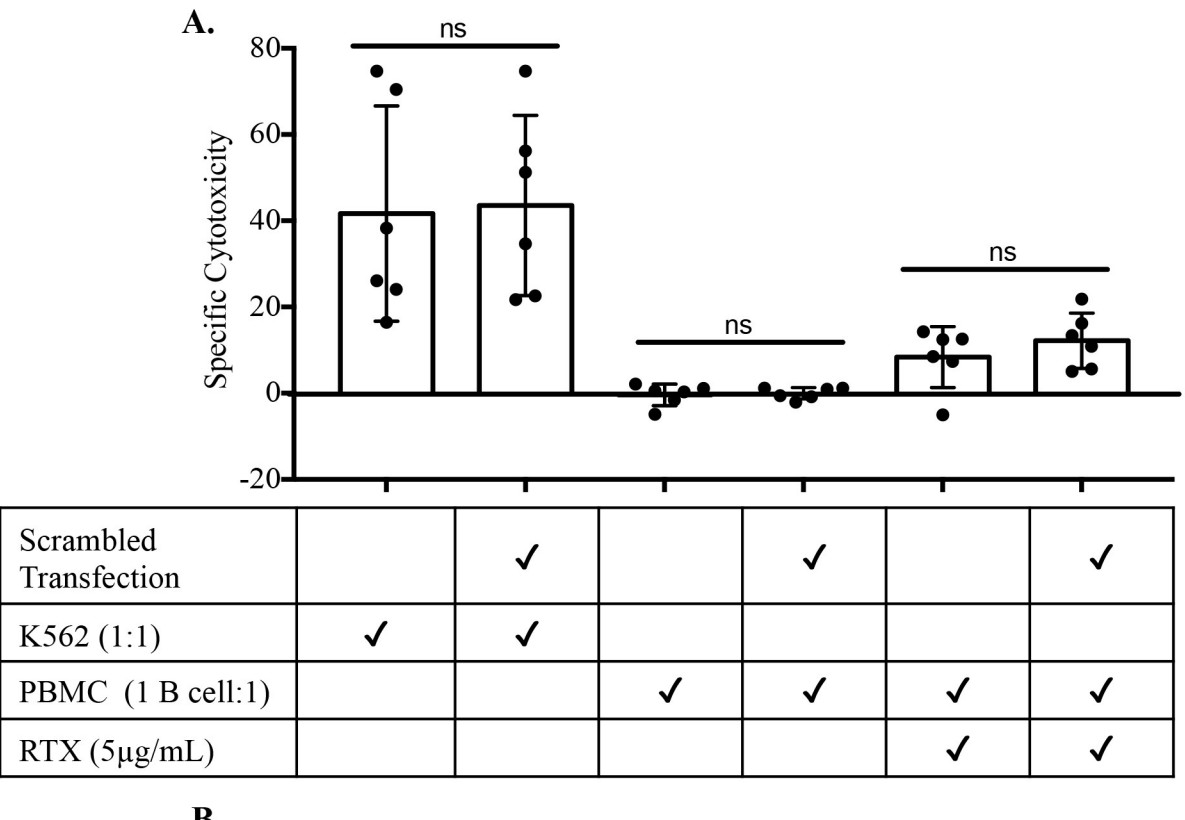

| | | | | | | |
|---|---|---|---|---|---|---|
| Scrambled Transfection | | ✓ | | ✓ | | ✓ |
| K562 (1:1) | ✓ | ✓ | | | | |
| PBMC (1 B cell:1) | | | ✓ | ✓ | ✓ | ✓ |
| RTX (5µg/mL) | | | | | ✓ | ✓ |

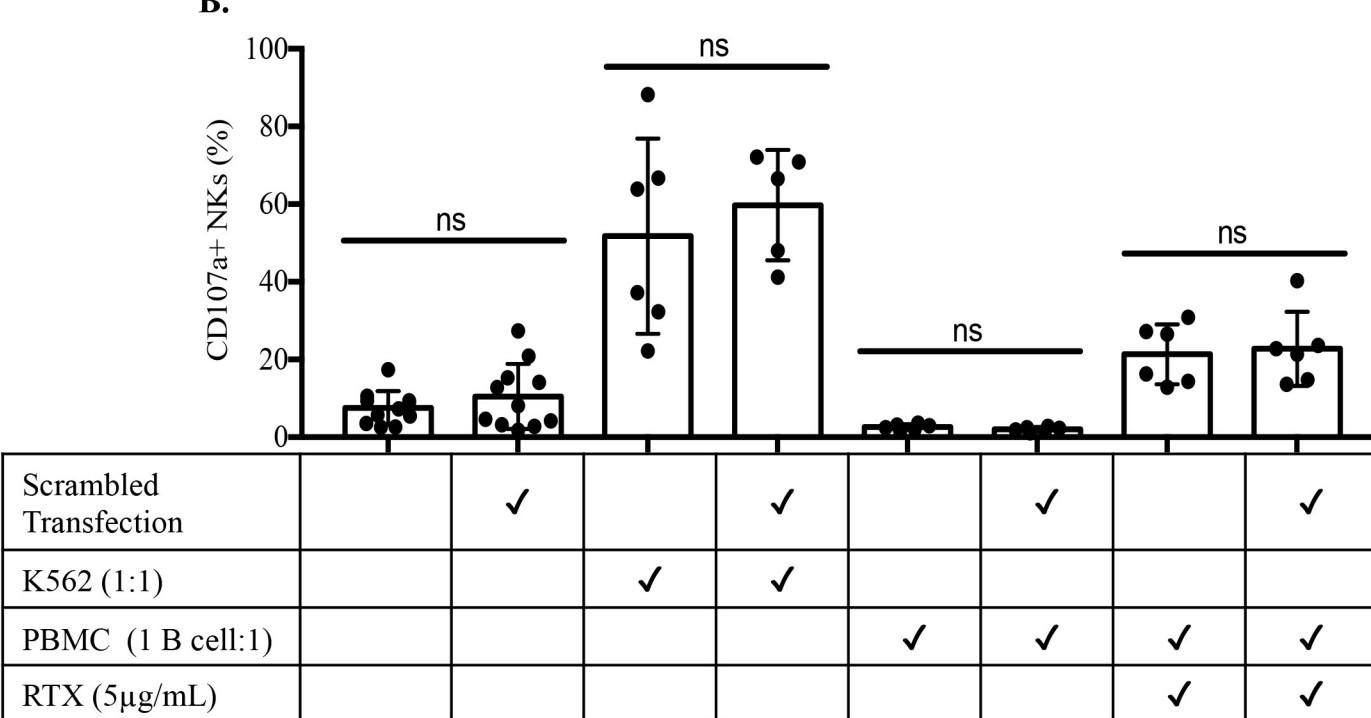

| | | | | | | | | |
|---|---|---|---|---|---|---|---|---|
| Scrambled Transfection | | ✓ | | ✓ | | ✓ | | ✓ |
| K562 (1:1) | | | ✓ | ✓ | | | | |
| PBMC (1 B cell:1) | | | | | ✓ | ✓ | ✓ | ✓ |
| RTX (5µg/mL) | | | | | | | ✓ | ✓ |

**Fig 5. Scrambled oligonucleotide transfection has no effect on NK cell function.** Rosettesep isolated human NK cells were transfected with scrambled oligonucleotide using TransIT-TKO methodology and incubated for 2.5 days. Transfected and non-transfected NK cells were cultured alone, with K562 cells (1:1) or with autologous PBMCs such that the B cell to NK cell ratio was 1:1 with and without RTX (anti-CD20). **A)** Targeted cell death was assessed by flow cytometry. **B)** NK cells degranulation was assessed by surface expression of CD107a and was measured by flow cytometry. Dots represent individual healthy donors, bar graphs represent mean ± SD. Data was assessed by two way ratio paired *t* tests.

## Discussion

MicroRNA are presenting as important and complex controllers of immune function, particularly in NK cells. Studying miRNA is challenging because of their numerous and heterogeneous impacts, but understanding them may open the possibility of miRNA-based therapy or NK cell manipulation for control of cellular processes. Research on miRNA has been bottlenecked by a lack of ideal approaches to alter the miRNA landscape without off-target effects. We report that both primary NK cells and the NK-92 cell line can be efficiently transfected for alterations of miRNA in serum-free conditions. Cellular viability, phenotype and function are maintained for at least four days in cell culture, and NK cells exhibit increases or decreases in specific miRNA expression after introduction of sense or antisense miRNA, respectively. Using a negative miRNA mimic control, we show that miRNA introduction and the use of TransIT-TKO alone do not alter NK cell function for missing-self reactivity or ADCC, or significantly impact the distribution of NK cell populations. Taken together, this work represents a novel and highly efficient technique for manipulating the miRNA environment in NK cells that will facilitate studies of miRNA impacts on NK cells.

In this study, we focused on two miRNA species with known impacts on NK cell function: miR-146a-5p and miR-155-5p. These species are generally associated with negative and positive feedback on NK cell activation, respectively. MiRNA-146a-5p is activated through NFκB and inhibits pro-inflammatory responses, including production of type I IFN, IL-1β and TNF, by regulating TRAF6 and STAT1; this in turn regulates NK cell function and maturation [38, 39]. In contrast, miR-155-5p is most commonly associated with pro-inflammatory features, including increasing production of IFN-γ and TNF production [40]. Here, we demonstrate that the expression of these miRNA species can be altered within 18h of transfection, with impacts on expression of mRNA for their target genes. Given the pleiotropic and dynamic nature of miRNA programs, altering them may provide broad changes in immunologic programs that would be beneficial in therapeutic applications or informative in disease processes.

Capable of immune polarization, target cell killing and self-tolerance, NK cells are attractive as tools for immunotherapy. NK cells do not require strict HLA matching, and have not been associated with substantial toxicity or off-target effects and may therefore be applicable as off-the-shelf adoptive cell therapies [22, 41, 42]. Increasingly, there is interest in engineering NK cells (i.e. chimeric antigen receptors or cytokines), for fit-for-purpose applications in immunotherapy [21]. Despite their advantages NK cell-based clinical trials have yielded heterogeneous results, likely reflecting further obstacles, such as immunosuppression in a tumor microenvironment, that hinder their reactivity [43, 44]. Analysis of miRNA transcriptomes from NK cells derived from peripheral blood, cord blood, and uterine decidua reveal significant differences in miRNA profiles, suggesting that miRNA may help to calibrate and tailor NK cell effector functions. Manipulation of miRNA profiles may complement ongoing engineering by allowing control of intracellular networks to buffer environmental signals and allow persistent NK cell migration, reactivity or regulation [18].

The high stability of miRNA, and that it is released and available in body fluids make it attractive candidates as biomarkers or for targeted therapies [16–18]. Although transfection techniques are numerous, efficient transfection of primary NK cells without introducing off-target effects or skewing the NK cell population remains challenging. In NK cells, transfection is associated with TLR activation, reduced NK cell function and apoptosis, as NK cell pattern recognition receptors respond to components of the delivery vector or the oligonucleotides themselves [23, 45]. Viral transfection can efficiently and permanently deliver a transgene of interest, but typically requires activation and/or proliferation of NK cells, and repeated infections and long culture times to be effective [23]. Noteworthy, lentiviral transduction requires

ongoing cellular division [46], which is incompatible with understanding the impacts of miRNA on resting NK cells. These infections and long culture times can have unintended consequences, convey insertional mutagenesis or alter the distribution or function of NK cell subsets inadvertently, and require strict safety precautions [45, 47]. Hence, for short-term study or alteration of miRNA in primary NK cells, non-viral transfections are desirable.

In our hands, only TransIT-TKO could efficiently transfect primary NK cells with high efficiency without substantial loss of viability. To our knowledge, the introduction of miRNA to NK cells using the TransIT TKO system represents the most efficient transfection method for alteration of miRNA in primary NK cells, outcompeting other methods of oligonucleotide delivery, including non-viral transfection methods. In our analysis, we assessed markers both of NK cell education (i.e. KIRs, NKG2A, LIR-1) and adaptation/memory (i.e. NKG2C, CD57, CD2, PLZF) [8, 9], finding that none are substantially altered through transfection alone. This opens the possibility to study the role(s) of miRNA or clusters of miRNA in NK and other cell types without substantial alterations from their resting state.

The magnitude of targeted miRNA reduction with use of miRNA inhibitors was less than that of the increase prompted by transfection of sense miRNA. This could reflect that these anti-sense miRNAs may bind to and inhibit miRNA translational repression of mRNA without initiating the miRNA degradation. Due to limitations in the viability and persistence of NK cells *in vitro*, we were unable to measure the impacts or persistence of transfected miRNA mimics or inhibitors beyond 2.5 days in these experiments. However, as miRNA are thought to act relatively quickly to repress mRNA in cells (i.e. 12-24h) [48], a 2.5-day window may nonetheless be sufficient to study the impact of miRNA on NK cell function.

NK cells are potent immune effectors already known to be important for controlling the outcomes of pregnancy, infection and cancer. Distinct NK cell subsets drive inflammation or impact immunoregulatory functions, and restrict adaptive immune responses that may otherwise lead to autoimmunity [49–51]. Variation in NK cell function within and between individuals is known to correspond with diversity in health outcomes, making this population highly interesting for study and precise immunotherapy. Our approach for genetic manipulation of primary human NK cells that introduces minimal changes in NK cell function, phenotype or viability, and can be completely conducted using serum-free media, presenting the possibility of manipulating NK miRNA to understand its roles, and to tailor NK cell function.

## Supporting information

**S1 Fig. X-VIVO serum free media best supports NK cell growth for cellular transfections.** NK-92 cells (Top) and RosetteSep isolated primary human NK cells (Bottom) were grown in ATCC recommended media (containing 25% serum) and serum free X-VIVO media for up to 4 days. **A/C)** Images represent magnification with 20x objective lens. **C/D)** Cellular viability were assessed by trypan blue exclusion. Data represents individual (B) or mean (D) values ± standard deviation, n = 1–3. Data was assessed by *t* test, ns indicates no significance. (DOCX)

**S2 Fig. TransIT-TKO is effective for transfection of lymphocytes and fibroblasts.** RosetteSep isolated primary human NK cells (**A**) primary human PBMCs (**B**) and primary JIA FLS (**C**) were transfected with miR-146a-5p sense or antisense miRNA compared to non-transfected control cells. **A-C)** Cellular viability, purity, and efficiency were determined by flow cytometry. **D)** MiRNA delivery was assessed by RTqPCR. Baseline reflects expression level of mi-146a-5p in cells transfected with negative control miRNA. Fold changes compared to negative were calculated using two reference miRNAs and the Pflaff Method. Data represents individual measurements and bars represent mean ± standard deviation, n = 1–3. RTqPCR results

were assessed by one-way ratio paired *t* tests.
(DOCX)

**S1 Table. Qiagen miRCURY LNA sense and antisense miRNA sequences.**
(DOCX)

**S2 Table. Primer sequences, efficiencies, and annealing temperatures for miRNA.**
(DOCX)

**S3 Table. Primer sequences, efficiencies, and annealing temperatures for mRNA.**
(DOCX)

**S4 Table. Flow cytometry antibodies, dyes and labels.**
(DOCX)

**S5 Table. Non-exhaustive MiRBase sequence blast.**
(DOCX)

## Acknowledgments

We sincerely thank the healthy blood donors for their contribution to this project and gratefully acknowledge the Dalhousie University Faculty of Medicine Flow Cytometry CORE facility, Derek Rowter, Renee Raudonis and Fang Liu for technical assistance with this project.

## Author Contributions

**Conceptualization:** Breanna K. V. Hargreaves, Beata Derfalvi, Jeanette E. Boudreau.

**Data curation:** Breanna K. V. Hargreaves, Sarah E. Roberts, Beata Derfalvi, Jeanette E. Boudreau.

**Formal analysis:** Breanna K. V. Hargreaves, Sarah E. Roberts, Beata Derfalvi, Jeanette E. Boudreau.

**Funding acquisition:** Beata Derfalvi, Jeanette E. Boudreau.

**Investigation:** Breanna K. V. Hargreaves, Sarah E. Roberts, Beata Derfalvi, Jeanette E. Boudreau.

**Methodology:** Breanna K. V. Hargreaves, Sarah E. Roberts, Beata Derfalvi, Jeanette E. Boudreau.

**Project administration:** Beata Derfalvi, Jeanette E. Boudreau.

**Resources:** Beata Derfalvi, Jeanette E. Boudreau.

**Supervision:** Beata Derfalvi, Jeanette E. Boudreau.

**Validation:** Breanna K. V. Hargreaves, Sarah E. Roberts, Beata Derfalvi, Jeanette E. Boudreau.

**Visualization:** Sarah E. Roberts, Beata Derfalvi, Jeanette E. Boudreau.

**Writing – original draft:** Breanna K. V. Hargreaves, Beata Derfalvi, Jeanette E. Boudreau.

**Writing – review & editing:** Sarah E. Roberts, Beata Derfalvi, Jeanette E. Boudreau.

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
