## [Decision Letter · Decision Letter 0]

4 Feb 2020

PONE-D-20-01243

High efficiency serum-free manipulation of miRNA in human NK cells without loss of viability or phenotypic alterations

PLOS ONE

Dear Dr. Boudreau,

Thank you for submitting your manuscript to PLOS ONE. After careful consideration, we feel that it has merit but does not fully meet PLOS ONE’s publication criteria as it currently stands. Therefore, we invite you to submit a revised version of the manuscript that addresses the points raised during the review process.

Please respond to all critique, Point-by-Point. In particular:

Transfection efficiencies were determined by fluorescence and qRT-PCR. Efficient transfection should be verified by the modulation of target gene expression, such as STAT1, as suggested by reviewer 1.

The discrepancy between the text (Results, line 268) and Figure 4, KIR2L2/L3 should be resolved, as requested by reviewer 2.

We would appreciate receiving your revised manuscript by Mar 20 2020 11:59PM. To enhance the reproducibility of your results, we recommend that if applicable you deposit your laboratory protocols in protocols.io, where a protocol can be assigned its own identifier (DOI) such that it can be cited independently in the future. For instructions see: http://journals.plos.org/plosone/s/submission-guidelines#loc-laboratory-protocols

We look forward to receiving your revised manuscript.

Kind regards,

Klaus Roemer

Academic Editor

PLOS ONE

Journal Requirements:

Reviewers' comments:

Reviewer's Responses to Questions

**Comments to the Author**

1. Is the manuscript technically sound, and do the data support the conclusions?

Reviewer #1: Partly

Reviewer #2: Partly

2. Has the statistical analysis been performed appropriately and rigorously? 

Reviewer #1: Yes

Reviewer #2: Yes

3. Have the authors made all data underlying the findings in their manuscript fully available?

Reviewer #1: No

Reviewer #2: Yes

4. Is the manuscript presented in an intelligible fashion and written in standard English?

Reviewer #1: Yes

Reviewer #2: Yes

5. Review Comments to the Author

Reviewer #1: Reading the title and abstract of this paper, a report on a new miRNA transfection method efficient in primary NK cells is expected. Instead, the aim of this paper is a comparative analysis of existing methods for the transient transfection of miRNA in primary NK cells. In particular, the authors compare the efficiency of different methods and evaluate side effects on cell viability.

The topic is interesting due to the known difficulties in obtaining an efficient transfection and an acceptable viability of primary NK cells.

However, the paper shows critical limitations that can be summarized in four main points.

1. The title and abstract should be modified to clearly summarize the real content of the paper

2. The authors consider only four transfection methods, two of which of the same producer.

A systematic description of the currently available methods is completely absent.

The authors should at least briefly describe the different kinds of methods motivating their choice of limiting their analysis to the four ones described.

3.The transfection efficiency is evaluated by the use of fluorescently-labeled miRNAs and by real-time PCR. It is known that part of the delivered miRNA can be functionally inactive (Thomson et al. Plos One 2013, 8: e55214. https://doi.org/10.1371/journal.pone.0055214). Consequently, miRNA transfection efficiency should be evaluated by a miRNA functional assay.

4. The authors should test the expression modulation induced by a specific miRNA on a known target gene. For example, in NK cells STAT1 is a documented target of miR-146a, which authors used for transfection experiments, (Xu et al. Cell Mol Immunol 2017, 14: 712-720). STAT1 mRNA expression should be detected in primary NK cells transfected with a miR-146a mimic/inhibitor or with a miRNA negative control.

Minor points

Table 1 should be included in the Supplementary Data.

Line 210, Supplementary Table 4, not 3

Lane 226, “Each of our transfectants…” not clear, the authors should simply describe that they transfected a fluorescently-labeled negative control miRNA mimic

Lines 232-238, Authors should describe in the text that they transfected FAM-labeled miR-146 mimic and inhibitor in the described cell types.

Lines 241-248, Authors should better describe the results reported in Fig.2, which are partly obtained by flow cytometry, partly by real-time PCR.

Reviewer #2: The study describes the establishment of a protocol for transfection of micro RNA into primary NK cells. In the rationale, the scientific value of the study is well described and the results obtained are correctly ranked. The individual chapters are written in an understandable way and the procedure is easy to follow. The result of the study is a new and very efficient method to introduce miRNA into primary NK cells without off-target effects. This opens up the possibility of improving the activity of NK cells in a targeted manner and thus optimising the potential of NK cells in the context of adaptive transfer of effector cells. Thus, the study has a sufficient impact on this field.

However, before this study can be published, some additions or improvements should be made to the manuscript.

Major revisions:

Results / line 243: Two miRNAs (155-5p and 146a-5p) were used in this study to determine the transfection efficiency under different conditions, with a „known impact on the function of NK cells“. I think it is urgently necessary to describe this "impact" more detailed in the corresponding chapter on methods.

Results / line 246: The authors write that transfection with both anti-sense miRNAs led to a significant decrease in the number of copies of the respective sense miRNA in the NK cells. In the corresponding figure 2B, however, this decrease is only significant for anti-sense miRNA-146a-5p. Although it is obvious that the transfection of anti-sense miRNA-155-5p also led to a reduced number of the corresponding sense-miRNA in the transfected NK cells, the result has to be described in this way.

Results / line 249 ff.: For the subsequent experiments to further characterize the transfection with TransIT-TKO, a "mimic" miRNA was used to exclude influences of the transfected miRNA on the phenotype and activity of the successfully transfected NK cells. This also made perfect sense. However, it would then be necessary to mention the transfection efficiency of the mimic miRNA and the NK-cell viability after transfection and ideally to show the corresponding figures in the supplementary materials.

However, if the transfection efficiency for mimic miRNA is significantly lower than that of NK cell-relevant miRNA (155-5p or 146a-5p), the validity of control transfections using mimic miRNA to investigate transfection-related side effects on NK cells would be questionable.

Results / line 268: In line 268/269 it is written that the so-called educated NK cells under K562 stimulation achieved higher IFN-� production and degranulation than their uneducated counterparts. However, the corresponding figure (Figure 4 / KIR2L2/L3) certainly does not show a significant difference between these two NK-cell types with regard to their target-induced activity. The authors should therefore substantiate this assertion with statistical numbers or change their statement accordingly.

Remark: As described in line 118 in the chapter "Cell lines and culture", 100 U/ml IL-2 were used for the culture of NK cells. Thus, this cytokine was dosed very highly. The effector cells used in the study were therefore actually lymphokine-activated killer cells (LAK). Whether this high dose had any effect on miRNA transfection remains open. However, the results of the ADCC tests were certainly influenced by this. Due to the fact, that all controls were applied with the same medium, it can be assumed that the results and conclusions derived from the assays are nonetheless correct. I would like to ask the authors for a short statement outside the manuscript.

Minor revisions

Generally: The authors should pay attention to a continuous identical designation of the miRNA (miR-155-5a / miRNA-155-5p).

Abstract / line 26: Although it is not yet clear to what extent which miRNA interacts with its mRNA via decradation and/or inhibition of translation, it is clear that miRNA does not affect the "transcription" of mRNA. Generally speaking, miRNA interferes with the "expression" of the respective mRNA.

Methods / line 126 It is certainly not mandatory to report the exact concentration of "NEAA" and "penicillin/streptomycin" here. But writing "1X" is of no use to anyone.

Methods / line 145 Were 100 or 110 U IL-2/ml used?

Methods / line 153 „Sense (mimic)“ would be the better spelling.

Methods / line 175 If it’s not a name, the spelling of „FcBlock“ is unusual (Fc block).

Methods / line 193 Why were the K562 co-cultures incubated for 5 hours and the PBMC co-cultures for only 2 hours?

Results / line 253 e. g. „phenotype“ instead of „function“ would be correct in this context.

Results / line 254 See commentary in line 253.

Figure 1 / legend Please describe which miRNA (miRNA-155-5p or miRNA-146a-5p) was used for the data displayed in the correspondimg figures.

Why is the number of volunteers whose NK cells were transfected using TranIT-SiQuest lower than those whose NK cells were transfected with lipofectamine or TansIT-TKO, respectively?

Figure 2 / Y-axis As a characteristic of quantitative PCR, the designation of the Y-axis in figure 2A with "��Cq" is definitely wrong and on the same axis in figure 2B at least not absolutely necessary, because this axis was sufficiently labelled with "Fold Change".

Figure 4 / legend + Scrambled Oligonucleotide

6. PLOS authors have the option to publish the peer review history of their article (what does this mean?). If published, this will include your full peer review and any attached files.

Reviewer #1: No

Reviewer #2: No

---

## [Author Response · Author response to Decision Letter 0]

20 Mar 2020

Response to the editor: 

E1. Transfection efficiencies were determined by fluorescence and qRT-PCR. Efficient transfection should be verified by the modulation of target gene expression, such as STAT1, as suggested by reviewer 1.

Response: This is indeed an important consideration. To demonstrate that miRNA transfected in this way are functional, we have appended a new panel to Figure 2 and explained in-text that reductions in the mRNA for two miRNA-146a-5p targets are observed at 18h post-transfection. 

E2. The discrepancy between the text (Results, line 268) and Figure 4, KIR2L2/L3 should be resolved, as requested by reviewer 2.

Response: We have removed the link to NK cell education, although we do maintain that the average IFN-γ production among educated NK cells is, on average, higher than that of the uneducated population. However, additional samples would likely be needed in order to confidently draw conclusions about education since extensive inter-individual variation indeed occurs. With this in mind, we felt it best to maintain the legend indicating educated and uneducated populations for the interested reader, but refrain from drawing conclusions about NK cell education based on this data. 

Response to reviewer #1: 

1. The title and abstract should be modified to clearly summarize the real content of the paper

Response: We understand how the previous title may have been misleading, and have changed it to reflect the content of the paper, which reveals transIT-TKO as a highly effective approach for transfecting primary human NK cells: 

“Highly efficient serum-free manipulation of miRNA in human NK cells without loss of viability of phenotypic alterations is accomplished using TransIT-TKO”

2. The authors consider only four transfection methods, two of which of the same producer.

A systematic description of the currently available methods is completely absent.

The authors should at least briefly describe the different kinds of methods motivating their choice of limiting their analysis to the four ones described.

Response: Respectfully, the goal of this work was to identify an efficient protocol with which to transfect miRNA to NK cells without altering their phenotype or viability. We did not feel that it was necessary to exhaustively test all available protocols once we achieved >90% efficiency and viability without changes to NK cell phenotype. Nevertheless, we agree that it is worth acknowledging the approaches that have been used to date to modify miRNA expression and now include this as part of our introduction and discussion. Surprisingly few studies have attempted to alter primary human NK cells’ miRNA profile, and most do not report transfection efficiency. It is our sincere hope that our work will enable studies of miRNA impacts in primary human NK cells. 

To address this comment, we have added a paragraph to the introduction to describe the existing knowledge and published attempts at miRNA modification in primary human NK cells, noting that a major limitation of each is the inclusion of serum in the media. 

3…. miRNA transfection efficiency should be evaluated by a miRNA functional assay.

4. The authors should test the expression modulation induced by a specific miRNA on a known target gene... mRNA expression should be detected in primary NK cells transfected with a miR-146a mimic/inhibitor or with a miRNA negative control.

Response: Whether transfected miRNA is indeed functional is an important consideration. We now include data to demonstrate decreased mRNA expression for two targets of miRNA146a-5p (STAT-1 and IRAK1) 18h post-transfection (Figure 2C). 

Minor points

5. Table 1 should be included in the Supplementary Data.

Response: We have moved Table 1 to become Supplementary Table 2 and relabeled supplementary tables throughout the supplement and manuscript text. 

6. Line 210, Supplementary Table 4, not 3

Response: We apologize for this error and have corrected the supplementary table labeling throughout. This is now supplementary Table 5, after moving the original Table 1 to supplementary information and relabeling appropriately.

7. Lane 226, “Each of our transfectants…” not clear, the authors should simply describe that they transfected a fluorescently-labeled negative control miRNA mimic

Response: Since each of our transfected oligonucleotides (carrying a specific or scrambled payload) was labeled with FAM, we clarified this to read as follows: 

“We used a fluorescein (FAM)-labeled control miRNA which encodes only a “scramble” sequence (i.e. no specific miRNA) to compare transfection approaches. The FAM label was included in this and all transfections (control, mimic and antisense). FAM allowed us to track transfection efficiency as the proportion of FAM+ among viable NK cells after transfection, and persistence of labeled oligonucleotides.”

8. Lines 232-238, Authors should describe in the text that they transfected FAM-labeled miR-146 mimic and inhibitor in the described cell types.

Response: We apologize for the confusion and have clarified these statements. In fact, each transfectant (not only the control) was coupled to FAM. In Figure 1 (described in these lines), cells were transfected with a FAM-labeled control (scramble) miRNA, not a miR-146 mimic. This was included in the figure caption and the text has now been updated to make this clear. Furthermore, ALL of our transfectants (control, mimic or antisense miRNA) were labeled with FAM. To clarify, we updated this statement to read as follows: 

“We used a fluorescein (FAM)-labeled control miRNA which encodes only a “scramble” sequence (i.e. no specific miRNA) to compare transfection approaches. The FAM label was included in this and all transfections (control, mimic and antisense transfectants) and allowed us to track transfection efficiency as the proportion of FAM+ among viable NK cells after transfection, and persistence of labeled oligonucleotides.”

9. Lines 241-248, Authors should better describe the results reported in Fig.2, which are partly obtained by flow cytometry, partly by real-time PCR.

Response: Thank you for this critique. The text has now been updated to reflect that data were collected by flow cytometry (panel A) and RT-qPCR (panel B). 

Response to reviewer #2: 

Major revisions:

1. Results / line 243: Two miRNAs (155-5p and 146a-5p) were used in this study to determine the transfection efficiency under different conditions, with a „known impact on the function of NK cells“. I think it is urgently necessary to describe this "impact" more detailed in the corresponding chapter on methods.

Response: We agree that demonstrating a functional impact of miRNA transfection is important and we have now included data to demonstrate how miRNA-146-5p diminishes mRNA for two of its targets, STAT1 and IRAK1 (results). In addition, we added a paragraph in the discussion to describe the important and opposing impacts of these two miRNA in NK cell biology. Respectfully, we felt it more appropriate to include these details in the the results and discussion sections than the methods section. 

2. Results / line 246: The authors write that transfection with both anti-sense miRNAs led to a significant decrease in the number of copies of the respective sense miRNA in the NK cells. In the corresponding figure 2B, however, this decrease is only significant for anti-sense miRNA-146a-5p. Although it is obvious that the transfection of anti-sense miRNA-155-5p also led to a reduced number of the corresponding sense-miRNA in the transfected NK cells, the result has to be described in this way.

Response: Thank you for bringing this to our attention. We have now updated the text to reflect that the change in miR-155-5p is a trend (i.e. lower, but not statistically significant). 

3. Results / line 249 ff.: For the subsequent experiments to further characterize the transfection with TransIT-TKO, a "mimic" miRNA was used to exclude influences of the transfected miRNA on the phenotype and activity of the successfully transfected NK cells. This also made perfect sense. However, it would then be necessary to mention the transfection efficiency of the mimic miRNA and the NK-cell viability after transfection and ideally to show the corresponding figures in the supplementary materials. However, if the transfection efficiency for mimic miRNA is significantly lower than that of NK cell-relevant miRNA (155-5p or 146a-5p), the validity of control transfections using mimic miRNA to investigate transfection-related side effects on NK cells would be questionable.

Response: We confirm that transfection efficiencies are similar among all treatments, including the scramble oligonucleotide controls. That the transfection efficiencies were similar between each treatment has now been explicitly stated in the results section of our manuscript.

3. Results / line 268: In line 268/269 it is written that the so-called educated NK cells under K562 stimulation achieved higher IFN-g production and degranulation than their uneducated counterparts. However, the corresponding figure (Figure 4 / KIR2L2/L3) certainly does not show a significant difference between these two NK-cell types with regard to their target-induced activity. The authors should therefore substantiate this assertion with statistical numbers or change their statement accordingly.

Response: This is an important critique. We have softened the language around this assertion. By comparing the educated (open) and uneducated (filled) circles, we respectfully assert that there is a difference based on education within this population (Educated: 34.0 +/-16.7%; Uneducated: 16.0 +/- 11.4%). However, we recognize that making this assertion on small groups is premature. Hence, we have removed the commentary about educated versus uneducated cells from the text of our work, but leave intact the legends to indicate the education status of each donor. 

4. Remark: As described in line 118 in the chapter "Cell lines and culture", 100 U/ml IL-2 were used for the culture of NK cells. Thus, this cytokine was dosed very highly. The effector cells used in the study were therefore actually lymphokine-activated killer cells (LAK). Whether this high dose had any effect on miRNA transfection remains open. However, the results of the ADCC tests were certainly influenced by this. Due to the fact, that all controls were applied with the same medium, it can be assumed that the results and conclusions derived from the assays are nonetheless correct. I would like to ask the authors for a short statement outside the manuscript.

Response: Thank you for this critique. Inclusion of IL-2 in cell culture media is critical for maintaining NK cell viability. Therefore it is indeed a limitation of this study that our NK cells may be better represented as LAK cells. We can confirm that all studies were completed in the same (optimized) media, and maintain that our results are therefore internally controlled for the impact of IL-2 exposure. We did not expressly titrate the cytokine to the lowest possible dose, but do note that similar concentrations of IL-2 are used in other studies of NK cell function with multi-day in vitro maintenance (i.e. Lauwerys et al., J Immunol 2000). Higher concentrations of IL-2 (500 IU/mL) can induce expression of miRNA species, including miRNA-155, but this is also true of IL-15 and IL-21, two other cytokines that may be chosen to support NK cells in vitro (Liu et al Immunology Letters 2012). Future studies will be required to minimize alterations in miRNA species that might occur as a result of culture conditions, and these might be best supported by multiple overlapping approaches to maintain NK cells in vitro, where the impacts of cytokine treatment and deliberate miRNA management can be ascertained. 

Minor revisions

5. Generally: The authors should pay attention to a continuous identical designation of the miRNA (miR-155-5a / miRNA-155-5p).

Response: We have combed through the manuscript and ensured that miR155-5p is written as such throughout the text, tables, supplement and figures. 

6. Abstract / line 26: Although it is not yet clear to what extent which miRNA interacts with its mRNA via decradation and/or inhibition of translation, it is clear that miRNA does not affect the "transcription" of mRNA. Generally speaking, miRNA interferes with the "expression" of the respective mRNA.

Response: We have changed “transcription” to “expression” in the abstract. 

7. Methods / line 126 It is certainly not mandatory to report the exact concentration of "NEAA" and "penicillin/streptomycin" here. But writing "1X" is of no use to anyone.

Response: We have removed the 1x from both NEAA and penicillin/streptomycin.

8. Methods / line 145 Were 100 or 110 U IL-2/ml used?

Response: 100 IU/mL IL-2 was used. Thank you for identifying this error; it has been corrected. 

9. Methods / line 153 „Sense (mimic)“ would be the better spelling.

Response: We have corrected this to now read: 

“negative control, mimic (sense) or antisense miRNA”

10. Methods / line 175 If it’s not a name, the spelling of „FcBlock“ is unusual (Fc block).

Response: This has been corrected to Fc Block

11. Methods / line 193 Why were the K562 co-cultures incubated for 5 hours and the PBMC co-cultures for only 2 hours?

Response: Antibody-mediated cellular cytotoxicity is a highly-efficient mechanism for killing. We found that 2h was sufficient to identify target cell lysis without background cytotoxicity. In contrast, while degranulation primarily occurs against K562 in the first 2h, we find that cytokine production takes slightly longer. In our hands, 5h was the ideal time to observe maximal IFN-γ production by NK cells. 

12. Results / line 253 e. g. „phenotype“ instead of „function“ would be correct in this context.

Results / line 254 See commentary in line 253.

Response: We agree. These substitutions have been made. 

13. Figure 1 / legend Please describe which miRNA (miRNA-155-5p or miRNA-146a-5p) was used for the data displayed in the corresponding figures.

Response: For these optimization experiments, we used a FAM-labeled scramble control oligo (i.e. a transfection control oligo). Neither miRNA-155-5p or miRNA-146a-5p was used. We have updated the caption text to read: 

“RosetteSep isolated primary human NK cells were transfected with 25 nM FAM-labeled negative control (scramble oligo) for 24 hours…”

14. Why is the number of volunteers whose NK cells were transfected using TranIT-SiQuest lower than those whose NK cells were transfected with lipofectamine or TansIT-TKO, respectively?

Response: As our goal was to identify an optimal approach to delivering miRNA to primary NK cells, we tested several donors with each approach. These experiments were limited by the number of NK cells that could be collected from our volunteer donors on a day-to-day basis, so not every technique could be tested with each collection. Once clear and significant patterns were detected, it was not necessary to analyze more donors. 

15. Figure 2 / Y-axis As a characteristic of quantitative PCR, the designation of the Y-axis in figure 2A with "DDCq" is definitely wrong and on the same axis in figure 2B at least not absolutely necessary, because this axis was sufficiently labelled with "Fold Change".

Response: We apologize for this obvious error and oversight and have corrected both problems in the resubmitted figure. 

16. Figure 4 / legend + Scrambled Oligonucleotide

Response: Corrected. Thank you.

---

## [Editor Report · Decision Letter 1]

30 Mar 2020

Highly efficient serum-free manipulation of miRNA in human NK cells without loss of viability or phenotypic alterations is accomplished with TransIT-TKO

PONE-D-20-01243R1

Dear Dr. Boudreau,

We are pleased to inform you that your manuscript has been judged scientifically suitable for publication and will be formally accepted for publication once it complies with all outstanding technical requirements.

With kind regards,

Klaus Roemer

Academic Editor

PLOS ONE
---

## [Editor Report · Acceptance letter]

7 Apr 2020

PONE-D-20-01243R1 

Highly efficient serum-free manipulation of miRNA in human NK cells without loss of viability or phenotypic alterations is accomplished with TransIT-TKO 

Dear Dr. Boudreau:

I am pleased to inform you that your manuscript has been deemed suitable for publication in PLOS ONE. Congratulations! Your manuscript is now with our production department. 

With kind regards,

on behalf of

Dr. Klaus Roemer 

Academic Editor

PLOS ONE